# Deep learning-based prediction of atrial fibrillation from polar transformed time-frequency electrocardiogram

**Daehyun Kwon**[☯]**, Hanbit Kang**[☯]**, Dongwoo Lee, Yoon-Chul Kim**[iD]*

Medical Artificial Intelligence Laboratory, Division of Digital Healthcare, College of Software and Digital Healthcare Convergence, Yonsei University, Wonju, Republic of Korea

☯ These authors contributed equally to this work.
* yoonckim@yonsei.ac.kr

**Data Availability Statement:** The data are publicly available from the PhysioNet/CinC Challenge 2017 database (https://physionet.org/content/challenge-2017/1.0.0/).

## Abstract

Portable and wearable electrocardiogram (ECG) devices are increasingly utilized in healthcare for monitoring heart rhythms and detecting cardiac arrhythmias or other heart conditions. The integration of ECG signal visualization with AI-based abnormality detection empowers users to independently and confidently assess their physiological signals. In this study, we investigated a novel method for visualizing ECG signals using polar transformations of short-time Fourier transform (STFT) spectrograms and evaluated the performance of deep convolutional neural networks (CNNs) in predicting atrial fibrillation from these polar transformed spectrograms. The ECG data, which are available from the PhysioNet/CinC Challenge 2017, were categorized into four classes: normal sinus rhythm, atrial fibrillation, other rhythms, and noise. Preprocessing steps included ECG signal filtering, STFT-based spectrogram generation, and reverse polar transformation to generate final polar spectrogram images. These images were used as inputs for deep CNN models, where three pretrained deep CNNs were used for comparisons. The results demonstrated that deep learning-based predictions using polar transformed spectrograms were comparable to existing methods. Furthermore, the polar transformed images offer a compact and intuitive representation of rhythm characteristics in ECG recordings, highlighting their potential for wearable applications.

## 1. Introduction

Portable and wearable electrocardiogram (ECG) devices are increasingly utilized in healthcare for monitoring heart rhythms and detecting cardiac arrhythmias or other heart conditions [1]. These devices typically record single-lead ECG signals, which, while less comprehensive than the standard 12-lead ECG used in clinical settings, offer longer monitoring durations, enhancing their utility for detecting arrhythmic events [2]. Among cardiac arrhythmias, atrial fibrillation (Afib) is particularly important as Afib is one of the major causes of acute ischemic stroke due to thromboembolism [3]. Accurate identification of Afib is essential to guide appropriate

**Funding:** "This study was supported by the "Regional Innovation Strategy (RIS)" through the National Research Foundation of Korea (NRF) and funded by the Ministry of Education (MOE) (2022RIS-005)."

**Competing interests:** The authors have declared that no competing interests exist.

thrombolytic therapies, which differ from treatments for ischemic strokes caused by large-vessel atherosclerosis [4].

Machine learning techniques have been widely adopted to automate the classification of cardiac arrhythmias and other physiological abnormalities using ECG data [5–11]. Feature extraction from ECG data transformed using the tunable Q-factor wavelet transform (TQWT) [12, 13] and machine learning have been performed to detect sleep apnea [9]. Deep learning models, including recurrent neural networks (RNNs), long short-term memory (LSTM), and gated recurrent unit (GRU) networks, have been utilized to identify Afib directly from raw ECG signals without requiring extensive preprocessing [14–16]. Time-frequency representations, commonly used in processing one-dimensional time-series data such as speech and heartbeat sound signals, have also been applied to ECG signals. Spectrograms, which reveal temporal changes in frequency distribution, have been converted into 2D images and used as inputs for deep convolutional neural networks (CNNs) to classify both adult and fetal cardiac arrhythmias [17–19]. Furthermore, integrating ECG spectrograms with RNNs has shown promise for addressing longer-range dependencies inherent in ECG signals [20]. Variations of 2D time-frequency representations such as continuous wavelet transforms [21], constant-Q non-stationary Gabor transforms [22], and S-transforms [8], have also been investigated.

Recent studies have investigated the potential of polar mappings for representing ECG signals or multi-parametric data. For example, polar representations such as the iris-spectrogram transform rectangular ECG spectrograms into polar images, where time progresses azimuthally and frequency increases radially [23]. These iris-spectrograms, derived from single-heartbeat ECG signals, have been used with CNN models to classify cardiac arrhythmia, bradycardia, and tachycardia [24, 25]. Another study proposed polar representations that integrate clinical features with 24-hour Holter ECG recordings to predict heart failure stages using deep CNN classifiers [26].

In this study, we present a visual representation technique for identifying Afib and normal sinus rhythm using reverse polar transformation of ECG spectrograms. The motivation behind the use of polar transformation stems from the intuition that for the case of a long duration of ECG signal, conventional spectrograms appear long in the horizontal axis, and the polar representation can provide compact visualization in a square form. Unlike iris-spectrograms, which are generated from one-heartbeat ECG signals [25], our method transforms longer duration ECG signals (30 seconds) into reverse polar spectrograms. Our approach to reverse polar transformation leverages the observation that high energy in ECG spectrograms resides in the low-frequency range. By reversing the frequency axis, high-energy regions are moved toward the periphery of the polar image, enhancing the visual discrimination between Afib and normal sinus rhythm. These transformed images are then used as inputs to deep CNN classifiers for the prediction of Afib. The study evaluates multi-class prediction results, integrating ECG preprocessing schemes with deep CNN models.

The main contributions of this study are:

1. Novel reverse polar transformed visual representations of ECG time-frequency data are demonstrated for effective identification of Afib in single-lead ECG data.

2. Polar transformed ECG spectrogram images are utilized as inputs to deep CNN models for cardiac arrhythmia classification.

3. The effectiveness of the Pan-Tompkins (P-T) preprocessing algorithm is demonstrated in enhancing deep CNN-based cardiac arrhythmia prediction.

Our paper is organized as follows. Section 2 details the ECG data preprocessing steps, polar transformation methods, deep learning model training and validation, and performance

evaluation. Section 3 presents a comparison of experimental results. Section 4 discusses the findings, implications, and future research directions. Finally, Section 5 summarizes the presented work and its key outcomes.

## 2. Methods

This section outlines the ECG dataset used in our study, the steps for ECG signal preprocessing and polar transformation, and the development and validation of the deep CNN models. The workflow of the proposed method is illustrated in Fig 1. The process begins with preprocessing ECG signals to generate time-frequency spectrograms. These spectrograms undergo polar coordinate transformation, which is subsequently mapped onto a Cartesian grid to generate polar spectrogram images. These transformed images are then fed into a deep CNN classifier for arrhythmia detection. The Python source code implemented for our study is available at https://sites.google.com/yonsei.ac.kr/yoonckim/research/dl-prediction-of-afib-from-polar-transformed-ecg-spectrogram.

### 2.1. Data

We considered publicly available ECG data provided by the PhysioNet/CinC Challenge 2017 (https://physionet.org/content/challenge-2017/1.0.0/) [27]. The dataset comprises single-lead

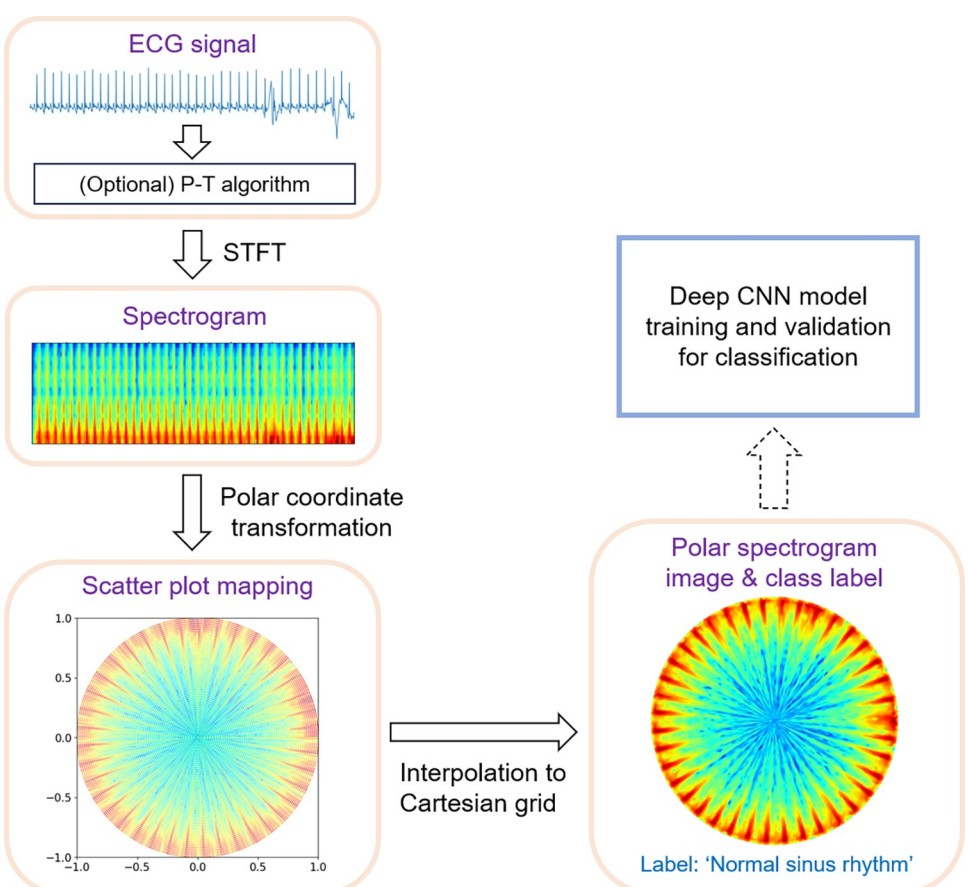

**Fig 1. Flowchart of the proposed polar spectrogram visualization method.** The polar spectrogram images and their corresponding labels are used to train and validate a deep CNN classification model. Abbreviations: ECG, electrocardiogram; P-T, Pan-Tompkins; STFT, short-time Fourier transform; CNN, convolutional neural network.

ECG recordings obtained using the AliveCor device (Mountainview, CA). The data consisted of 8,528 ECG recordings sampled at 300 Hz with a median duration of 30 seconds. Each recording was labeled into one of four classes: normal sinus rhythm, atrial fibrillation (Afib), other rhythm, or noise (too noisy to classify). The 'other rhythm' category encompasses arrhythmias such as premature ventricular contractions (PVCs), premature atrial contractions (PACs) and other abnormal rhythms excluding Afib.

## 2.2. Preprocessing of ECG signals

To preprocess the ECG signals, we utilized the Pan-Tompkins (P-T) algorithm [28]. This method applies a series of low-pass, high-pass, and derivative filters to remove background noise and improve the detection of heartbeat frequencies. After preprocessing, the output retained the frequency content of the ECG signals while suppressing background noise irrelevant to QRS complex detection. The P-T algorithm is effective for identifying abnormal heart rhythms but has a limitation. It may lose signals within R-R intervals, potentially impacting the detection of certain heart conditions such as myocardial infarction or hypertrophic cardiomyopathy.

Following preprocessing, we computed the spectrogram of the ECG signals using short-time Fourier transform (STFT). The STFT $X[t, f]$ of the ECG signal $x[n]$ is described as follows.

$$X[t,f] = \sum_{n=0}^{N-1} x[n]w[t-n]e^{-i2\pi fn} \tag{1}$$

Here, $w[t-n]$ represents the window function, which shifts by the amount of $t$ along the $n$-axis, generating time-localized frequency information.

To enhance visual clarity, we applied a logarithmic transformation of the STFT output to produce improved spectrograms. The spectrograms were computed using the stft function in Python's SciPy library [29]. Each segment length was set to 64, with 32 samples overlapping between segments. The nfft value was set to 128. The resulting spectrogram dimensions were 128 x 600.

## 2.3. Polar transformation

The 2D spectrograms were then mapped to polar coordinates. The polar transformation is mathematically described as follows.

$$P[x, y] = P[f \cos \theta, f \sin \theta) = X[t, f] \tag{2}$$

where $f$ corresponds to the vertical axis of the spectrogram $X[t, f]$, and the angle $\theta$ is given by:

$$\theta = 2\pi \cdot \frac{t}{T} \tag{3}$$

Here, $t$ ranges from 0 to $T-1$, corresponding to the horizontal axis of the spectrogram. As shown in Fig 1, during the transformation, unfilled spaces were observed in the scatter plot mapping, particularly near the periphery of the polar coordinate space. Inspired by the gridding technique commonly used in magnetic resonance imaging (MRI) [30, 31], we applied linear interpolation to fill these gaps and obtain polar spectrogram images on Cartesian grids. The resulting polar spectrograms exhibited higher intensity in the low-frequency regions, which were densely spaced (see Figs 2C and 3C). To improve discrimination between Afib and normal sinus rhythm, we performed a reverse polar transformation, which shifts high-energy

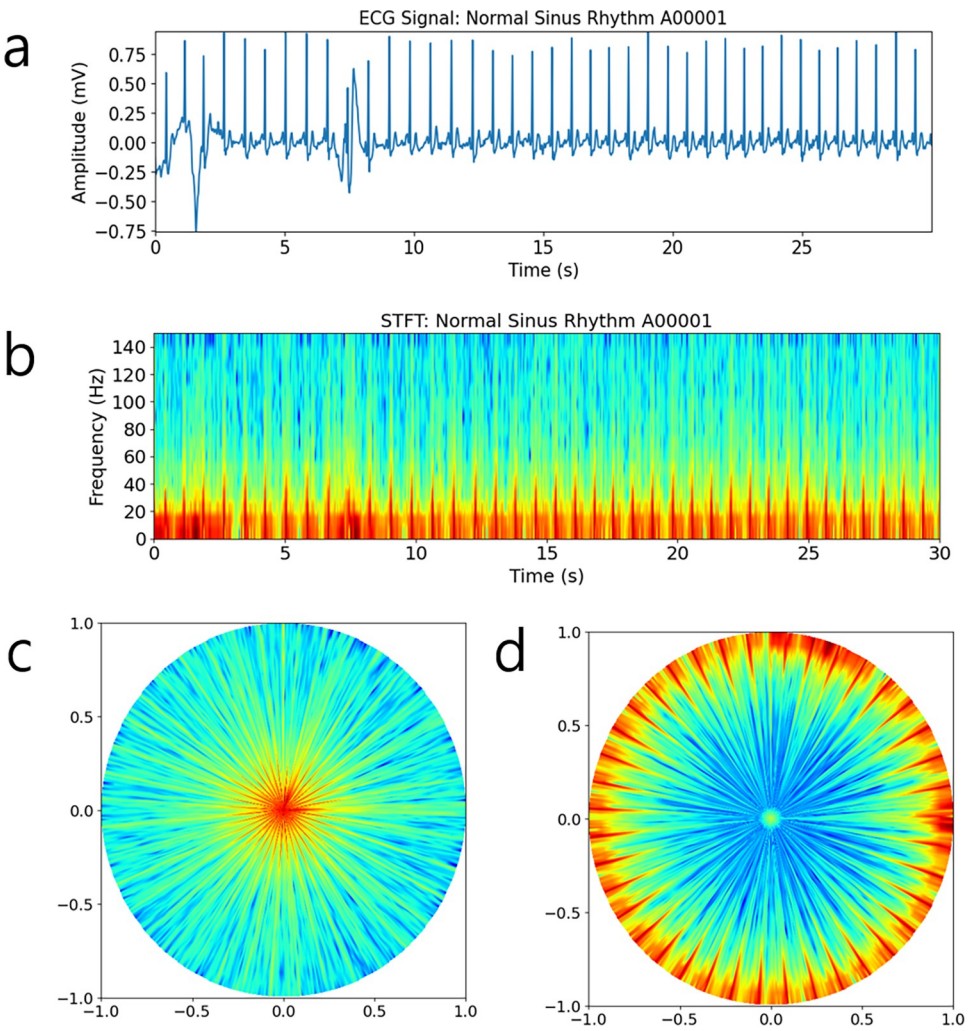

**Fig 2. Processed results from non-filtered ECG data.** (a) ECG time series, (b) spectrogram, (c) polar spectrogram, and (d) reverse polar spectrogram. Low frequency data is visualized in red, while high frequency data is visualized in blue. We used the 'jet' colormap for color display. Reverse polar transformed spectrogram images such as (d) were used to train, validate, and test the deep CNN models in our study.

regions toward the periphery of the polar space (see Figs 2D and 3D). The reverse polar transformation is mathematically described as follows.

$$P_{\text{rev}}[x, y] = P[(f_{\max} - f)\cos \theta, (f_{\max} - f)\sin \theta] \tag{4}$$

where $f_{\max}$ represents the maximum frequency of the spectrogram $X[t, f]$.

The reverse polar transformation enhances the visual distinction of non-uniformly spaced R-R intervals, aiding in Afib classification. High-energy components are relocated to the periphery of the polar space, making Afib signals more distinguishable from normal sinus rhythms.

The polar transformed images were colorized using the 'jet' colormap, which represents a color spectrum from blue to red. These images were resized to 224 x 224 pixels, their intensity values rescaled to the range [0, 255] as 8-bit unsigned integers, and saved in png format for subsequent deep learning model development and validation.

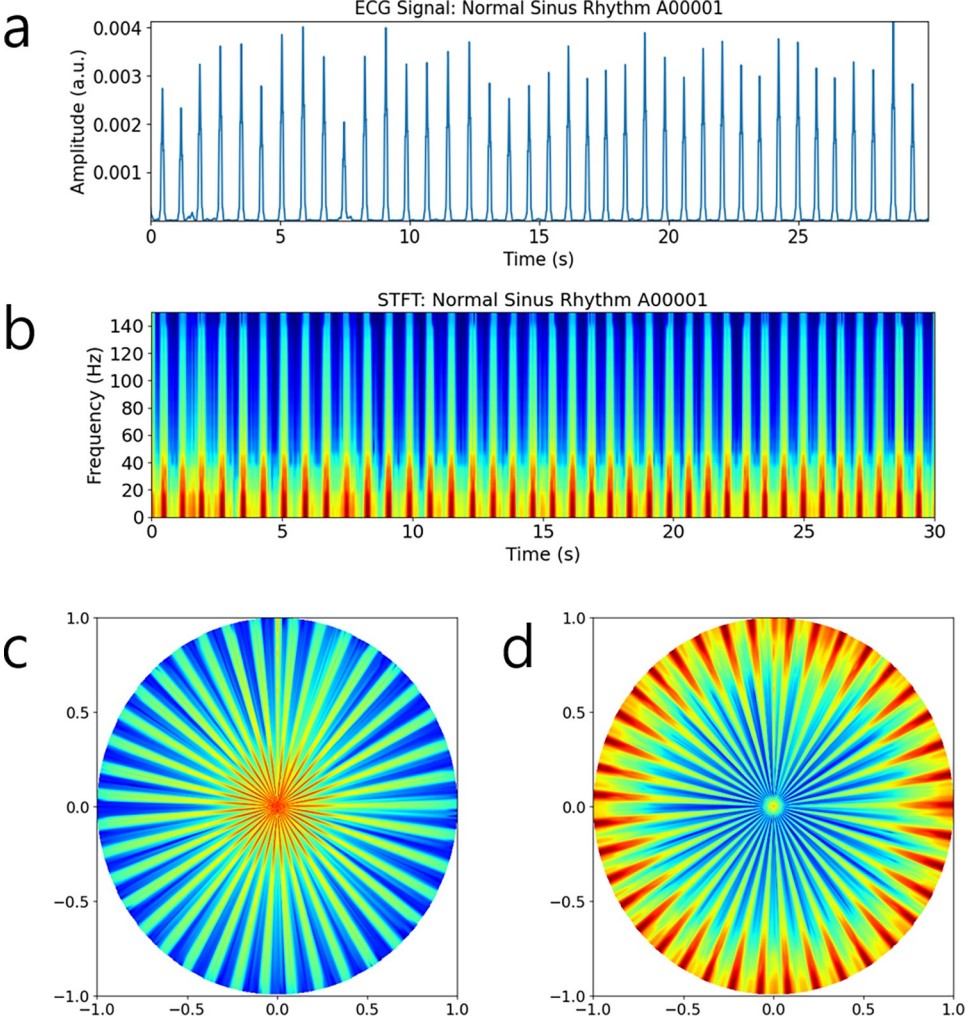

**Fig 3. Processed results from ECG data with the P-T algorithm.** (a) ECG time series after applying the P-T algorithm, (b) spectrogram, (c) polar spectrogram, and (d) reverse polar spectrogram. Reverse polar transformed spectrogram images such as (d) were used to train, validate, and test the deep CNN models in our study.

## 2.4. Model development

This subsection describes the development of deep learning models. Out of a total of 8,528 ECG recordings, 5,977 ECG recordings with a duration of 30 seconds were selected for training, validation, and testing. The polar transformed spectrograms from 4,781 ECG recordings were assigned to the model development group, while the remaining 1,196 recordings were designated as the test group. Table 1 lists the number of records per class for both groups.

**Table 1. The number of records for the model development and test data.**

| Class | Model development | Test |
| --- | --- | --- |
| Atrial fibrillation (Afib) | 409 | 90 |
| Normal sinus rhythm | 2,924 | 754 |
| Other rhythm | 1,352 | 323 |
| Noise | 96 | 29 |
| Total | 4,781 | 1,196 |

Despite the class imbalance issue, we did not utilize any data augmentation techniques. Five-fold cross-validation was performed within the model development group to train and validate five deep CNN models. Code for the model development was implemented in Keras [32]. Pre-trained models, including MobileNet [33], ResNet50 [34], and DenseNet121 [35], trained on the ImageNet dataset [36], were used as baseline architectures for feature extraction, with their weights frozen during training. Extracted features underwent global average pooling (GAP) [37] followed by a fully connected layer, outputting predictions for one of four classes: atrial fibrillation (Afib), normal sinus rhythm, other rhythm, and noise. The models were trained using the Adam optimizer [38] with a sparse categorical cross-entropy loss function. Accuracy metrics were evaluated for each training and validation epoch. Input images were resized to 224 x 224 x 3 to match the default input dimensions for Keras deep learning models. After experimenting with various learning rates, the optimal value was set to 0.001. Training and validation were performed for up to 50 epochs with model weights saved after each epoch. For each fold, the epoch yielding the highest validation accuracy was selected for final evaluation.

## 2.5. Performance evaluation

The deep CNN models were trained on a Windows PC equipped with 12th Gen Intel® Core™ i9-12900K, 32 GB RAM, and NVIDIA GeForce RTX 3080 GPU (10.0 GB memory). Two pre-processing schemes were evaluated:

1. Polar transformation of the spectrogram of raw ECG signals.

2. Polar transformation of the spectrogram of P-T preprocessed ECG signals.

   We tested four classification approaches:

a. Model using MobileNet as the baseline network.

b. Model using ResNet50 as the baseline network.

c. Model using DenseNet121 as the baseline network.

d. A voting classifier that combined predictions from the three baseline models.

   For models A-C, the final prediction was determined by a hard vote across the classification results of all five folds. The voting classifier D was implemented as follows. First, for each fold, a soft vote was performed using probability scores from the three models (MobileNet, ResNet50, DenseNet121). Second, a hard-vote aggregated the soft-vote results from all five folds. Since each preprocessing scheme involved four classification approaches, a total of eight methods were evaluated.

   Performance metrics, including F1-score, precision, recall, and accuracy, were calculated using the Scikit-learn library. For each class $c$, precision ($P_c$), recall ($R_c$), and F1-score ($F1_c$) were defined as follows.

$$P_c = \frac{TP}{TP + FP} \tag{5}$$

$$R_c = \frac{TP}{TP + FN} \tag{6}$$

$$F1_c = \frac{2 \cdot P_c \cdot R_c}{P_c + R_c} = \frac{2 \cdot TP}{2 \cdot TP + FN + FP} \tag{7}$$

where TP, FN, and FP are the true positives, false negatives, and false positives for class $c$,

respectively. For the multi-class classification problem, we adopted macro-averaging for evaluation. Since the noise class was excluded for the calculation of F1-score according to the CinC challenge 2017 guidelines, the macro F1-score (F1) was computed as follows.

$$F1 = \frac{F1_A + F1_N + F1_O}{3} \tag{8}$$

where $F1_A$, $F1_N$, and $F1_O$ represent the F1-scores for Afib, normal sinus rhythm, and other rhythm classes, respectively. The accuracy score was calculated as:

$$Acc = \frac{\text{Sum of correctly identified predictions for each class}}{\text{total number of samples}} \tag{9}$$

## 3. Results

This section presents a qualitative comparison between polar transformed images and our proposed reverse polar transformed images, as well as visualization results for raw and P-T preprocessed ECG signals. Quantitative results of deep CNN predictions on test data are also detailed, using various pretrained models. The interpretation of deep CNN's prediction results is based on reverse polar transformed spectrogram images.

Figs 2 and 3 compare ECG signals, spectrograms, polar spectrograms, and reverse polar spectrograms, both with and without P-T signal preprocessing. The P-T processed ECG signal exhibits flat zero amplitude between R-R intervals as shown in Fig 3A. Corresponding spectrograms (Fig 3B) reveal clearer separation of R-R intervals compared to spectrograms without preprocessing (Fig 2B). Polar transformed spectrograms shown in Figs 2C and 3C exhibit frequency increasing proportionally with distance from the origin. This dense spacing of red rays may obscure arrhythmic events. In contrast, reverse polar spectrograms in Figs 2D and 3D exhibit frequency decreasing with distance from the origin, improving visual clarity for identifying arrhythmic events due to the sparse spacing of peripheral red lines.

Table 2 summarizes the prediction performance of different deep CNN models on test data. The P-T preprocessing algorithm significantly improved performance across all metrics. For example, the MobileNet model with P-T preprocessing achieved a macro F1-score of 0.8012, compared to 0.6995 without preprocessing. The DenseNet121 model with P-T preprocessing achieved the highest macro F1-score (0.8238) and accuracy (0.9076), surpassing even the voting method. Due to class imbalance (Table 1), accuracy scores ranged from 0.8259 to 0.9076, higher than macro F1-scores (0.6384–0.8238). DenseNet121 with P-T preprocessing ranked highest across multiple metrics: F1-scores for normal sinus rhythm and other rhythm classes,

**Table 2. Prediction results on test data.**

| Neural network model | ECG filtering method | F1$_A$ | F1$_N$ | F1$_O$ | Macro F1-score | Macro precision | Macro recall | Accuracy |
|---|---|---|---|---|---|---|---|---|
| MobileNet | No | 0.6536 | 0.8823 | 0.5626 | 0.6995 | 0.7827 | 0.6590 | 0.8585 |
| | P-T* | 0.8068 | 0.9001 | 0.6967 | 0.8012 | 0.7786 | **0.8375** | 0.8943 |
| ResNet50 | No | 0.5079 | 0.8635 | 0.5439 | 0.6384 | 0.6467 | 0.6370 | 0.8259 |
| | P-T | 0.7607 | 0.9052 | 0.7069 | 0.7909 | 0.8299 | 0.7617 | 0.8931 |
| DenseNet121 | No | 0.4918 | 0.8906 | 0.6697 | 0.6840 | 0.8189 | 0.6458 | 0.8656 |
| | P-T | 0.8132 | **0.9179** | **0.7402** | **0.8238** | 0.8374 | 0.8156 | **0.9076** |
| Voting[†] | No | 0.6624 | 0.8937 | 0.6395 | 0.7318 | 0.7774 | 0.6987 | 0.8690 |
| | P-T | **0.8144** | 0.9076 | 0.7079 | 0.8100 | **0.8488** | 0.7841 | 0.8993 |

[†] Voting classifier combines predictions from MobileNet, ResNet50, and DenseNet121 using soft votes within each fold and hard votes across folds.

*: P-T indicates the Pan-Tompkins algorithm.

**Table 3. Comparison with existing methods.**

| | Input data type | Classifier | Macro F1-score |
|---|---|---|---|
| Rizwan et al. [39] | Features extracted from ECG signal | Decision tree ensemble | 0.80 |
| Warrick et al. [40] | 1D ECG signal | 1D CNN and LSTM layers | 0.82 |
| Zhao et al. [41] | Averaged spectrogram representations of 3 R-R duration ECG | DenseNet with 18 layers | 0.80 |
| Cao et al. [42] | Denoised ECG signal using wavelet transform | 2-layer LSTM | 0.82 |
| Cheng et al. [43] | Denoised ECG signal using a combination of wavelet transform and median filtering | 24-layer deep CNN and Bidirectional LSTM | 0.89 |
| Ours | Polar time-frequency representation of ECG signal | DenseNet-121 | 0.82 |

macro F1-score, and accuracy. The voting method with P-T preprocessing ranked highest in F1-score for Afib and macro precision, while MobileNet with P-T preprocessing achieved the highest macro recall.

Table 3 compares the proposed method with existing methods in the literature. Our method demonstrates comparable performance. However, note that our method did not utilize the hidden dataset available from the CinC challenge 2017 and only used the 30-seconds ECG dataset.

Fig 4 compares confusion matrices for test data predictions using P-T preprocessing. All four models appear to have produced similar prediction results. For the noise class having smaller test samples than the other three classes, from the confusion matrices the voting classifier shows the highest F1-score of 0.6667, which is higher than the F1-scores of 0.4390, 0.4762, and 0.5957 for the DenseNet121, ResNet50, and MobileNet models, respectively. Therefore, the voting classifier seems effective for the prediction of the noise class, which was not considered for the calculation of macro F1-score in our study.

Fig 5 presents t-Distributed Stochastic Neighbor Embedding (t-SNE) [44] visualizations of the penultimate feature distributions. Averaged penultimate features from the five cross-validated models were used as input to TSNE's fit_transform() function in Scikit-Learn [45]. The distributions indicate that in well-performing models (e.g., MobileNet with P-T preprocessing in Fig 5D), intra-class samples are more tightly clustered compared to less effective models (e.g., ResNet50 without P-T preprocessing in Fig 5B).

Fig 6 shows representative examples of correct predictions. In the examples, the MobileNet with P-T was used to predict one of the four classes. When comparing Fig 6A and 6B, it is clear that 'Afib' is characterized by wide gaps with blue colors along the azimuthal angle in a couple of the spokes indicating arrhythmia. Fig 6C also shows other arrhythmia characterized by more repetitive arrhythmic patterns.

Fig 7 displays representative examples of incorrect predictions. Fig 7A shows repetitive arrhythmia patterns which appear similar to Fig 6C. Hence, the deep learning model predicted 'Other' although the label was 'Afib'. Fig 7B shows that the deep learning model did a reasonable job when the human annotator labeled it as 'Normal'. Fig 7C shows very uniformly spaced spokes, so it would be easy to classify it as 'Normal', which was the output predicted by the deep learning model. However, it was labeled as 'Other', since the dense spacing of the spokes may indicate the sign of tachycardia. Fig 7D shows several vague lines of the heartbeats, indicating the 'Noise' class, but the deep learning model prediction was 'Afib'.

## 4. Discussion

To the best of our knowledge, this study presents the first demonstration of reverse polar-transformed spectrograms used as input to a deep CNN model for detecting atrial fibrillation

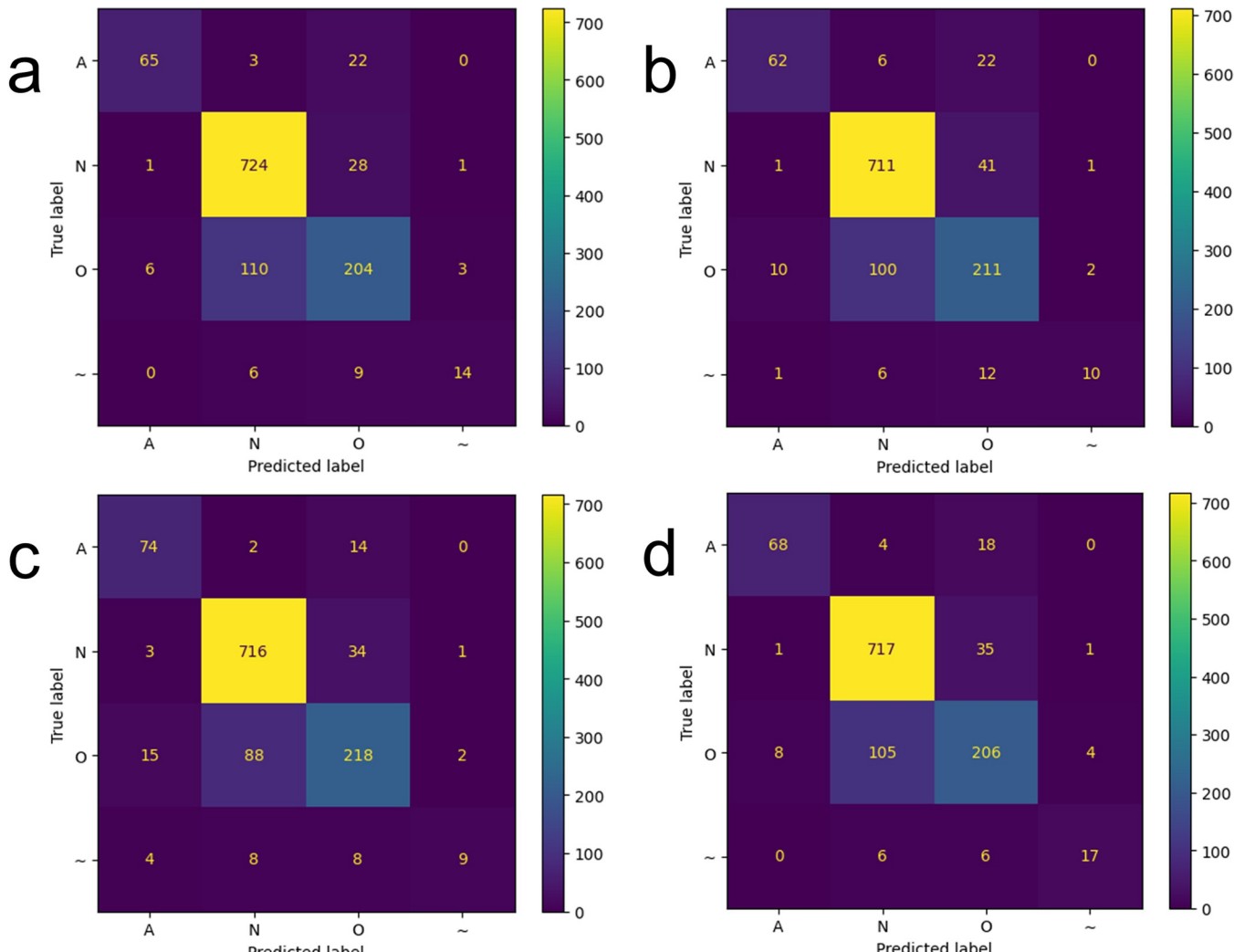

**Fig 4.** Confusion matrices for (a) MobileNet, (b) ResNet50, (c) DenseNet121, and (d) Voting classifiers when the P-T algorithm was used for ECG data preprocessing. A: Afib, N: normal sinus rhythm, O: other rhythm, and ~: noise.

(Afib). We focused on visualizing a 30-second ECG spectrogram in a polar representation. In this method, high energy in the spectrogram, typically concentrated in the low-frequency range, is moved to the periphery. This approach enhances the ability to distinguish cardiac arrhythmias from normal sinus rhythms by improving the visual representation of the spectrogram. A minor limitation of the polar representation is the irregularity in the appearance of the rays around the positive x-axis, where the start and end signals meet. This issue becomes more prominent with shorter ECG durations, as seen in the irregular spacing near the positive x-axis of the polar spectrogram image.

Our method differs from the previously proposed 'iris-spectrogram' method in two key ways. First, our focus is on rhythm classification, which requires a longer ECG signal (e.g., 30 seconds), compared to beat classification, which typically uses one R-R interval (~ 1 second). Second, we employed reverse polar transformation to broaden the spacing between R-R intervals, facilitating better visualization and classification.

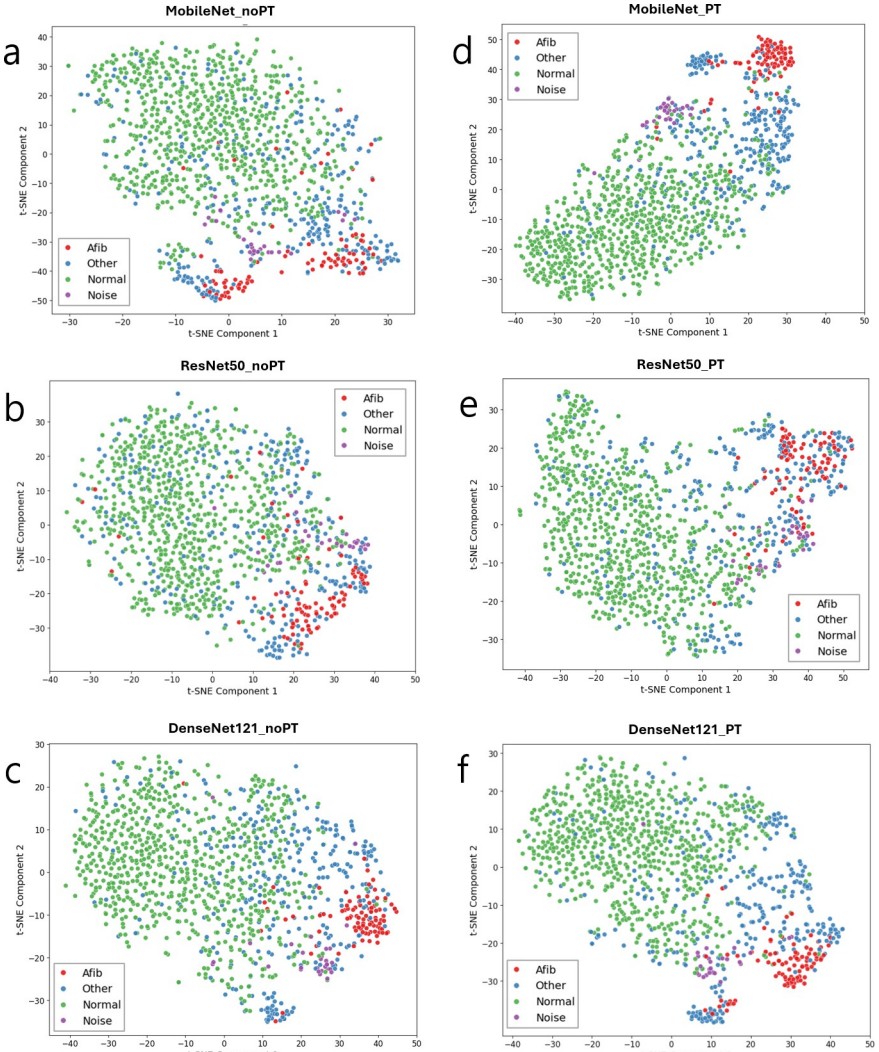

**Fig 5. tSNE visualization after dimensionality reduction of the penultimate features.** The test samples are shown in colors for difference classes. The left and right columns show results for without and with P-T preprocessing, respectively. The intra-class samples in the good-performance models (d-f) tend to be more clustered than those in the poor-performance models (a-c). Afib: atrial fibrillation, Normal: normal sinus rhythm, Other: other rhythm, Noise: noisy signal.

Preprocessing methods, such as the Pan-Tompkins (P-T) algorithm, showed improvements in deep learning model performance compared to unprocessed signals. However, the P-T algorithm suppresses segments of the ECG signal other than the R peaks, which could affect the identification of certain heart diseases. Despite this, the P-T algorithm proves useful in predicting arrhythmias, as arrhythmia identification is often based on irregular R peak patterns. Recent studies also highlight the potential of signal representations like the tunable Q-factor wavelet transform (TQWT) [46, 47] and discrete cosine transform [48] for improving arrhythmia detection and data compression, which could further enhance deep CNN predictions.

As ECG signal duration increases, it becomes challenging to display the full signal in conventional rectangular spectrograms. The polar representation addresses this by providing a complete view of the time-frequency pattern, making it more suitable for visualizing long-duration signals. This compact visualization could be particularly beneficial for mobile

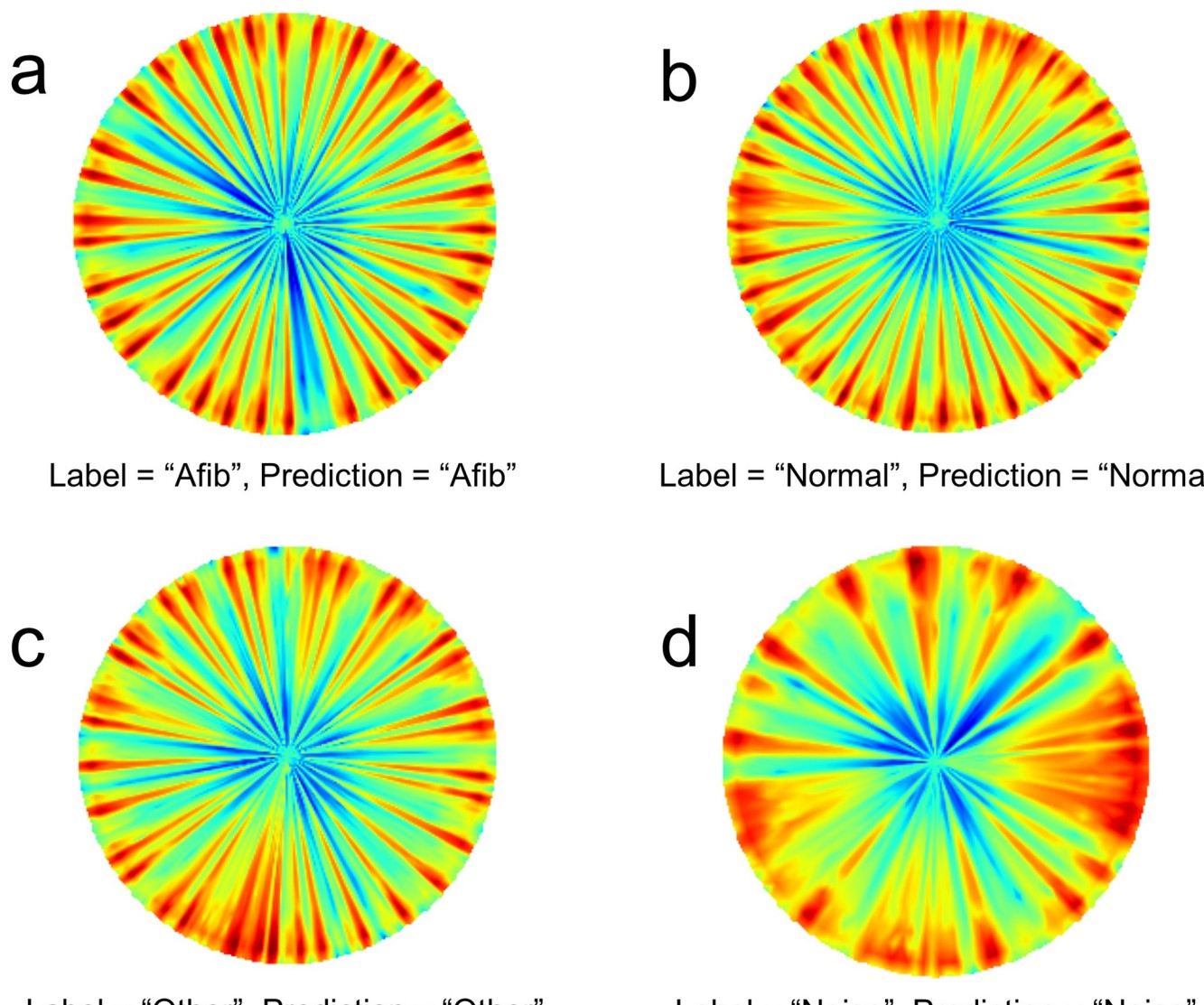

**Fig 6. Representative examples of correct predictions in the reverse polar transformed spectrograms.** Afib: atrial fibrillation, Normal: normal sinus rhythm, Other: other rhythm, Noise: noisy signal.

applications, where screen size is limited. Hence, polar transformed ECG spectrograms provide an efficient way to visualize long-duration time-series data on mobile devices.

The polar transformed images inherently have square matrix dimensions, which are ideal for input into deep CNN models, but it is noted that the azimuthal angle spacing is inversely proportional to the time duration of the ECG signal. Meanwhile, data augmentation techniques, such as random image rotations, could be applied to polar transformed spectrograms to diversify training datasets. Additionally, methods used for augmenting raw ECG signals [42] may be adapted for augmenting polar transformed spectrogram images. These data augmentation techniques may help improve the prediction performance.

Our method demonstrated performance comparable to the existing methods from the CinC 2017 challenge [49], with similar macro F1-scores (Table 3). However, this comparison is not entirely fair, as our approach only utilized ECG signals with a 30-second duration.

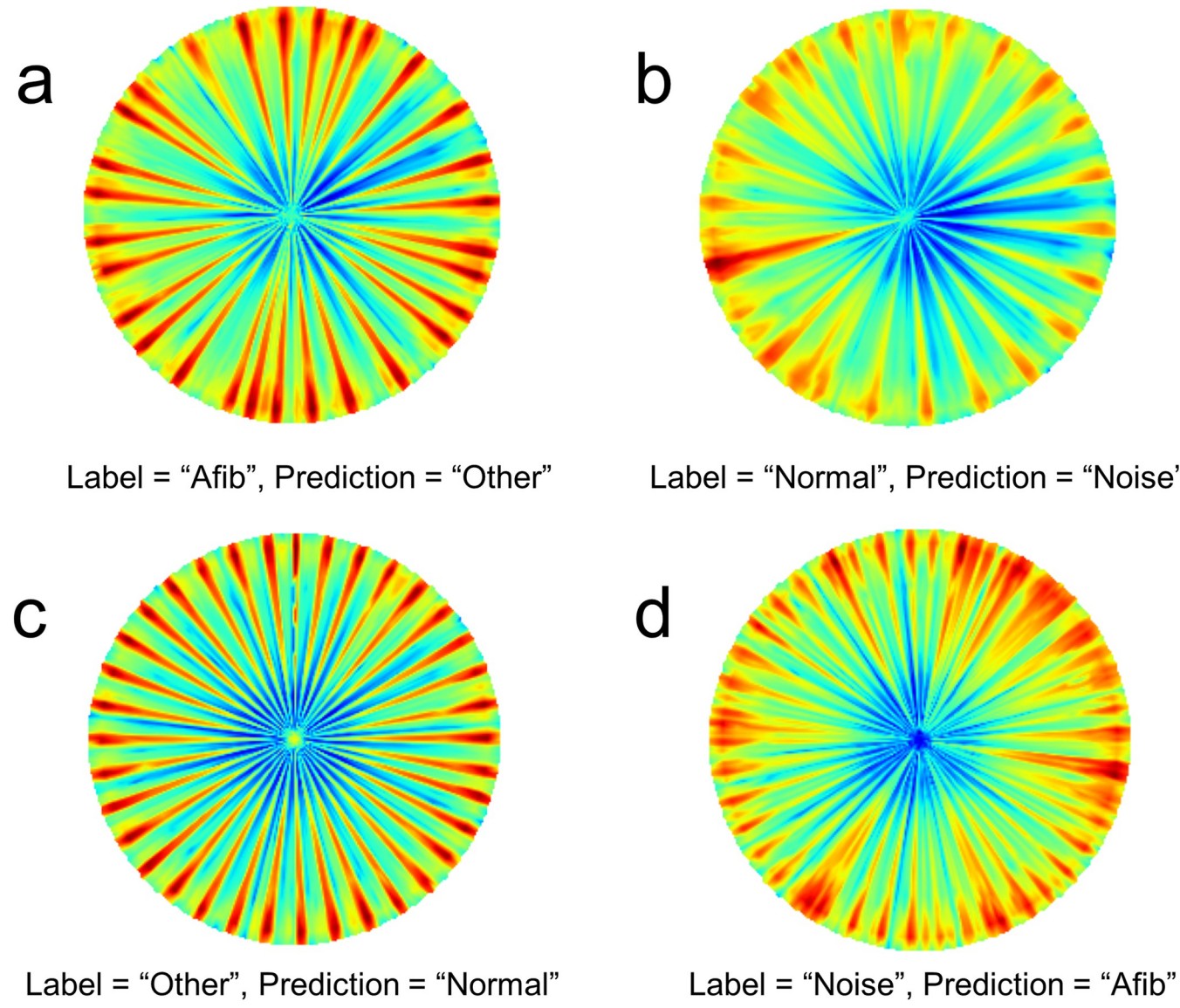

**Fig 7. Representative examples of incorrect predictions in the reverse polar transformed spectrograms.** Afib: atrial fibrillation, Normal: normal sinus rhythm, Other: other rhythm, Noise: noisy signal.

Future work will involve applying our method to any arbitrary duration of ECG signals or other publicly available ECG datasets [50] for further evaluation and validation.

## 5. Conclusions

This study introduces a novel reverse polar transformed spectrogram for visualizing 30-second ECG signals, which aids in the detection of atrial fibrillation (Afib). The reverse polar transformation enhances the visualization of cardiac arrhythmias by emphasizing irregular spacing between peaks at the periphery of the polar spectrogram. The reverse polar transformed spectrograms were successfully used as inputs for deep CNN models to predict Afib, achieving performance comparable to existing methods in the literature.

The proposed method offers advantages over traditional rectangular 2D spectrograms, such as compactly representing long-duration ECG signals and simplifying implementation due to its square matrix format, which is well-suited for widely used 2D CNN models. One limitation of the polar transformed spectrogram is its sensitivity to preprocessing filters and ECG signal amplitude, which can affect the color representation. Nevertheless, this approach holds promise for improving ECG signal abnormality detection and is particularly well-suited to Afib detection with existing popular deep CNN classifiers.

## Author Contributions

**Conceptualization:** Dongwoo Lee, Yoon-Chul Kim.

**Data curation:** Daehyun Kwon, Hanbit Kang.

**Formal analysis:** Daehyun Kwon, Hanbit Kang, Dongwoo Lee, Yoon-Chul Kim.

**Funding acquisition:** Yoon-Chul Kim.

**Investigation:** Hanbit Kang, Dongwoo Lee, Yoon-Chul Kim.

**Methodology:** Daehyun Kwon, Hanbit Kang, Dongwoo Lee, Yoon-Chul Kim.

**Project administration:** Yoon-Chul Kim.

**Resources:** Yoon-Chul Kim.

**Software:** Daehyun Kwon, Hanbit Kang, Dongwoo Lee.

**Supervision:** Yoon-Chul Kim.

**Validation:** Daehyun Kwon, Hanbit Kang, Dongwoo Lee, Yoon-Chul Kim.

**Visualization:** Daehyun Kwon, Hanbit Kang, Dongwoo Lee, Yoon-Chul Kim.

**Writing – original draft:** Daehyun Kwon, Hanbit Kang, Yoon-Chul Kim.

**Writing – review & editing:** Yoon-Chul Kim.

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
