## [Decision Letter · Decision Letter 0]

11 Jul 2024

PONE-D-24-23244Deep learning-based prediction of atrial fibrillation from polar transformed time-frequency electrocardiogramPLOS ONE

Dear Dr. Kim,

Thank you for submitting your manuscript to PLOS ONE. After careful consideration, we feel that it has merit but does not fully meet PLOS ONE’s publication criteria as it currently stands. Therefore, we invite you to submit a revised version of the manuscript that addresses the points raised during the review process.

We look forward to receiving your revised manuscript.

Kind regards,

Kapil Gupta, Ph.D

Academic Editor

PLOS ONE

“This study was supported by the “Regional Innovation Strategy (RIS)” through the National Research Foundation of Korea (NRF) and funded by the Ministry of Education (MOE) (2022RIS-005).”

6. Please amend either the abstract on the online submission form (via Edit Submission) or the abstract in the manuscript so that they are identical.

Additional Editor Comments:

Reviewer agree that the manuscript contains novel elements. However, it presents some aspects that need to be solved before reconsideration.

The authors should explicitly mention the significant contributions of the manuscript. The novelty of the paper is not highlighted.

The advantages and limitations of the proposed approach in relation to similar schemes are not clear.

Please revise the structure of the paper. It is recommendable to add to each section a couple of sentences that explain the purpose of the section. With this organization, the reader can clearly understand the sequence of the paper.

Under the review observations, the paper should be corrected as a major revision.

Reviewers' comments:

Reviewer's Responses to Questions

**Comments to the Author**

1. Is the manuscript technically sound, and do the data support the conclusions?

Reviewer #1: Yes

2. Has the statistical analysis been performed appropriately and rigorously? 

Reviewer #1: Yes

3. Have the authors made all data underlying the findings in their manuscript fully available?

Reviewer #1: Yes

4. Is the manuscript presented in an intelligible fashion and written in standard English?

Reviewer #1: No

5. Review Comments to the Author

Reviewer #1: This paper presents Deep learning-based prediction of atrial fibrillation. In general work is interesting although following issues are need to be resolve before ensuring recommendation.

1. The Research work is interesting but lags in terms of proving novelty in work.

2. Authors are required to write their contribution explicitly over the existing method. It seems that author has just used some methods and compared.

3. A little more mathematical analysis is required to support the proposed method.

4. References are need to be formed and updated properly as recently developed methods are not included in the literature work.

5. In this work, following recently published publications can be added

• Pal, H. S., Kumar, A., Vishwakarma, A., & Singh, G. K. (2023). Optimized Tunable-Q Wavelet Transform-Based 2-D ECG Compression Technique Using DCT. IEEE Transactions on Instrumentation and Measurement, 72, 1-13.

• Pal, H. S., Kumar, A., Vishwakarma, A., & Lee, H. N. (2023). Electrocardiogram signal compression using adaptive tunable-Q wavelet transform and modified dead-zone quantizer. ISA transactions, 142, 335-346.

• Pal, H. S., Kumar, A., Vishwakarma, A., Singh, G. K., & Lee, H. N. (2024). A new automated compression technique for 2D electrocardiogram signals using discrete wavelet transform. Engineering Applications of Artificial Intelligence, 133, 108123.

• Pal, H. S., Kumar, A., Vishwakarma, A., Singh, G. K., & Lee, H. N. (2024). An effective ECG signal compression algorithm with self controlled reconstruction quality. Computer Methods in Biomechanics and Biomedical Engineering, 27(7), 849-859.

• Gupta, K., Bajaj, V., & Jain, S. (2024). Multi-resolution assessment of ECG sensor data for sleep apnea detection using wide neural network. IEEE Sensors Journal.

• Gupta, K., Bajaj, V., & Ansari, I. A. (2023). Integrated s-transform-based learning system for detection of arrhythmic fetus. IEEE Transactions on Instrumentation and Measurement, 72, 1-8.

6. PLOS authors have the option to publish the peer review history of their article (what does this mean?). If published, this will include your full peer review and any attached files.

Reviewer #1: No

---

## [Author Response · Author response to Decision Letter 0]

6 Aug 2024

Manuscript ID: PONE-D-24-23244R1

Title: Deep learning-based prediction of atrial fibrillation from polar transformed time-frequency electrocardiogram

Response to Review

We thank the editor and reviewer for giving us the opportunity to revise this manuscript. Following the comments by the editor and reviewer, we have made significant changes to the manuscript. Listed below are responses to the comments along with descriptions of all changes to the manuscript. 

Authors’ response: As requested, we have modified our paper format to meet the style requirements. 

Authors’ response: We have shared our code to facilitate reproducibility in research. Our lab website at https://sites.google.com/yonsei.ac.kr/yoonckim/research/dl-prediction-of-afib-from-polar-transformed-ecg-spectrogram provides the Github links for the code. 

Authors’ response: We cannot modify the ‘Financial Disclosure’ section in the online submission site. Our Funding Statement should read as follows: 

“This study was supported by the “Regional Innovation Strategy (RIS)” through the National Research Foundation of Korea (NRF) and funded by the Ministry of Education (MOE) (2022RIS-005).” 

“This study was supported by the “Regional Innovation Strategy (RIS)” through the National Research Foundation of Korea (NRF) and funded by the Ministry of Education (MOE) (2022RIS-005).”

Authors’ response: We have removed the Acknowledgments section from the revised manuscript. Our Funding Statement should read as follows: 

“This study was supported by the “Regional Innovation Strategy (RIS)” through the National Research Foundation of Korea (NRF) and funded by the Ministry of Education (MOE) (2022RIS-005).” 

We have included our funding statement in our cover letter as well. 

Authors’ response: We have made the data available at our lab website at https://sites.google.com/yonsei.ac.kr/yoonckim/research/dl-prediction-of-afib-from-polar-transformed-ecg-spectrogram.

6. Please amend either the abstract on the online submission form (via Edit Submission) or the abstract in the manuscript so that they are identical.

Authors’ response: Thank you for pointing out the discrepancy. We ensured that they are identical in the revised manuscript submission. 

Additional Editor Comments: 

Reviewer agree that the manuscript contains novel elements. However, it presents some aspects that need to be solved before reconsideration.

1) The authors should explicitly mention the significant contributions of the manuscript. The novelty of the paper is not highlighted.

Authors’ response: We have provided the main contributions explicitly in the Introduction section of the revised manuscript. 

[Introduction section, page 5]

The main contributions of this study can be summarized as follows.

1. A novel reverse polar transformed visual representation of time-frequency ECG spectrogram is presented and demonstrated for the identification of Afib in single lead ECG data. 

2. The polar transformed ECG spectrogram images are used as input to a deep CNN model for the prediction of cardiac arrhythmia.

3. The effectiveness of the Pan-Tompkins (P-T) pre-processing algorithm is demonstrated for the prediction of cardiac arrhythmia when using deep CNNs.

2) The advantages and limitations of the proposed approach in relation to similar schemes are not clear.

Authors’ response: We have added the advantages and limitations to the Conclusions section.

[Conclusions section, page 16]

In sum, the proposed method is advantageous over the standard rectangular 2D representation with regard to its compact visualization of the long duration of ECG signal and its simplicity in implementation due to its square matrix form, which is suitable for existing and widely used 2D CNN models. The drawback of the proposed method in relation to the 1D ECG waveform analysis is that the color representation of the polar spectrogram is sensitive to the preprocessing filter and ECG signal amplitude.

3) Please revise the structure of the paper. It is recommendable to add to each section a couple of sentences that explain the purpose of the section. With this organization, the reader can clearly understand the sequence of the paper.

Authors’ response: We have added to each of the Methods and Results sections a couple of sentences that explain the purpose of the section. 

[2. Methods section, page 5] 

This section describes the ECG data used for our study, details of ECG signal pre-processing and polar transformation, and deep CNN model development and validation processes. The flowchart of the presented work is illustrated in Fig 1. ECG signals are processed to generate time-frequency spectrograms. After the polar coordinate transformation and mapping of it to the Cartesian grid, polar spectrogram images are generated, and then they are input to a deep CNN classifier model.

[2.4. Model development section, page 8]

This subsection describes the details of deep learning model development.

[3. Results section, page 11]

This section presents qualitative comparisons between polar transformed images and our proposed reverse polar transformed images. It also compares the visualization results between raw ECG signals and the P-T processed ECG signals. Quantitative results of deep CNN predictions on test data are shown across different CNN pre-trained models. The interpretation of deep CNN’s prediction results is made based on the reverse polar transformed spectrogram images.

4) Under the review observations, the paper should be corrected as a major revision.

Authors’ response: We sincerely thank the Editor for the decision. 

Reviewer comments:

Reviewer #1:

This paper presents Deep learning-based prediction of atrial fibrillation. In general work is interesting although following issues are need to be resolve before ensuring recommendation. 

Authors’ response: We greatly thank the reviewer for positive comments on our manuscript.

R1.1: The Research work is interesting but lags in terms of proving novelty in work. 

Authors’ response: The novelty is in the demonstration of a reverse polar spectrogram representation of a long duration of ECG signal and its application to the deep CNN-based prediction of atrial fibrillation. 

R1.2: Authors are required to write their contribution explicitly over the existing method. It seems that author has just used some methods and compared.

Authors’ response: We have provided the main contributions explicitly in the Introduction section of the revised manuscript. 

[Introduction section, page 5]

The main contributions of this study can be summarized as follows.

1. A novel reverse polar transformed visual representation of time-frequency ECG spectrogram is presented and demonstrated for the identification of Afib in single lead ECG data. 

2. The polar transformed ECG spectrogram images are used as input to a deep CNN model for the prediction of cardiac arrhythmia.

3. The effectiveness of the Pan-Tompkins (P-T) pre-processing algorithm is demonstrated for the prediction of cardiac arrhythmia when using deep CNNs.

R1.3: A little more mathematical analysis is required to support the proposed method.

Authors’ response: We have provided the mathematical description of the proposed transformation method in section 2.3. ‘Polar transformation’ of the revised manuscript on page 8 and 9. 

R1.4: References are need to be formed and updated properly as recently developed methods are not included in the literature work.

Authors’ response: We have added references that include recently developed methods. This is related to the response to R1.5. 

R1.5: In this work, following recently published publications can be added.

• Pal, H. S., Kumar, A., Vishwakarma, A., & Singh, G. K. (2023). Optimized Tunable-Q Wavelet Transform-Based 2-D ECG Compression Technique Using DCT. IEEE Transactions on Instrumentation and Measurement, 72, 1-13.

• Pal, H. S., Kumar, A., Vishwakarma, A., & Lee, H. N. (2023). Electrocardiogram signal compression using adaptive tunable-Q wavelet transform and modified dead-zone quantizer. ISA transactions, 142, 335-346.

• Pal, H. S., Kumar, A., Vishwakarma, A., Singh, G. K., & Lee, H. N. (2024). A new automated compression technique for 2D electrocardiogram signals using discrete wavelet transform. Engineering Applications of Artificial Intelligence, 133, 108123.

• Pal, H. S., Kumar, A., Vishwakarma, A., Singh, G. K., & Lee, H. N. (2024). An effective ECG signal compression algorithm with self controlled reconstruction quality. Computer Methods in Biomechanics and Biomedical Engineering, 27(7), 849-859.

• Gupta, K., Bajaj, V., & Jain, S. (2024). Multi-resolution assessment of ECG sensor data for sleep apnea detection using wide neural network. IEEE Sensors Journal.

• Gupta, K., Bajaj, V., & Ansari, I. A. (2023). Integrated s-transform-based learning system for detection of arrhythmic fetus. IEEE Transactions on Instrumentation and Measurement, 72, 1-8.

Authors’ response: We thank the reviewer for suggesting the references. We thought the suggested references are relevant to our work, and we cited them in the revised manuscript.

---

## [Decision Letter · Decision Letter 1]

20 Oct 2024

PONE-D-24-23244R1Deep learning-based prediction of atrial fibrillation from polar transformed time-frequency electrocardiogramPLOS ONE

Dear Dr. Kim,

Thank you for submitting your manuscript to PLOS ONE. After careful consideration, we feel that it has merit but does not fully meet PLOS ONE’s publication criteria as it currently stands. Therefore, we invite you to submit a revised version of the manuscript that addresses the points raised during the review process.

We look forward to receiving your revised manuscript.

Kind regards,

Hirenkumar Kantilal Mewada

Academic Editor

PLOS ONE

Additional Editor Comments:

The review has been received, and it indicates that major revisions are required. Please refer to the reviewer comments for specific areas needing improvement.

Reviewers' comments:

Reviewer's Responses to Questions

**Comments to the Author**

1. If the authors have adequately addressed your comments raised in a previous round of review and you feel that this manuscript is now acceptable for publication, you may indicate that here to bypass the “Comments to the Author” section, enter your conflict of interest statement in the “Confidential to Editor” section, and submit your "Accept" recommendation.

Reviewer #1: All comments have been addressed

Reviewer #2: (No Response)

2. Is the manuscript technically sound, and do the data support the conclusions?

Reviewer #1: Yes

Reviewer #2: (No Response)

3. Has the statistical analysis been performed appropriately and rigorously? 

Reviewer #1: Yes

Reviewer #2: (No Response)

4. Have the authors made all data underlying the findings in their manuscript fully available?

Reviewer #1: Yes

Reviewer #2: (No Response)

5. Is the manuscript presented in an intelligible fashion and written in standard English?

Reviewer #1: Yes

Reviewer #2: (No Response)

6. Review Comments to the Author

Reviewer #1: All the comments are addressed.

1. Improve the quality of figure 1-3 as fonts are not visible.

2. Update the references as per the journals guidelines.

3. some grammatical mistakes are need to be corrected, thus requires a thorough revision.

Reviewer #2: Discuss the motivation behind reverse polar transformed visual representation of time-frequency ECG spectrogram.

Deep learning models composed of the deep layers. By passing the ECG signal as an input directly to CNN model can predict or classify atrial fibrillation. For more details, authors can look at the article https://link.springer.com/chapter/10.1007/978-3-319-68385-0_18

The information behind transforming signal to an image is unclear.

This database is not a new one and there are many related works on CNN and other deep learning models. The proposed method results should be compared with at least existing 3 methods.

Discuss what are the advantages of the proposed method compared to the existing methods. Discuss the limitations of the proposed method.

Hidden layer feature visualization can be shown for example penultimate layer feature visualization using t-SNE and add discussion on this. Features also can be visualized using SHAP models. Discuss on these will support why the proposed model achieves better performances compared to the existing models.

7. PLOS authors have the option to publish the peer review history of their article (what does this mean?). If published, this will include your full peer review and any attached files.

Reviewer #1: No

Reviewer #2: **Yes: **Vinayakumar Ravi

---

## [Author Response · Author response to Decision Letter 1]

2 Dec 2024

Please see the attached file for the response to reviewers.

---

## [Decision Letter · Decision Letter 2]

2 Jan 2025

Deep learning-based prediction of atrial fibrillation from polar transformed time-frequency electrocardiogram

PONE-D-24-23244R2

Dear Dr. Kim,

We’re pleased to inform you that your manuscript has been judged scientifically suitable for publication and will be formally accepted for publication once it meets all outstanding technical requirements.

Kind regards,

Hirenkumar Kantilal Mewada

Academic Editor

PLOS ONE

Additional Editor Comments (optional):

Reviewers' comments:

Reviewer's Responses to Questions

**Comments to the Author**

1. If the authors have adequately addressed your comments raised in a previous round of review and you feel that this manuscript is now acceptable for publication, you may indicate that here to bypass the “Comments to the Author” section, enter your conflict of interest statement in the “Confidential to Editor” section, and submit your "Accept" recommendation.

Reviewer #2: All comments have been addressed

2. Is the manuscript technically sound, and do the data support the conclusions?

Reviewer #2: Yes

3. Has the statistical analysis been performed appropriately and rigorously? 

Reviewer #2: Yes

4. Have the authors made all data underlying the findings in their manuscript fully available?

Reviewer #2: Yes

5. Is the manuscript presented in an intelligible fashion and written in standard English?

Reviewer #2: Yes

6. Review Comments to the Author

Reviewer #2: Revised paper can be published

Authors addressed all comments.

Authors are suggested to check the journal guidelines

7. PLOS authors have the option to publish the peer review history of their article (what does this mean?). If published, this will include your full peer review and any attached files.

Reviewer #2: **Yes: **Vinayakumar Ravi

---

## [Editor Report · Acceptance letter]

7 Jan 2025

PONE-D-24-23244R2 

PLOS ONE

Dear Dr. Kim, 

I'm pleased to inform you that your manuscript has been deemed suitable for publication in PLOS ONE. Congratulations! Your manuscript is now being handed over to our production team.

Kind regards, 

on behalf of

Dr. Hirenkumar Kantilal Mewada 

Academic Editor

PLOS ONE